# Micropropagation of Endemic Endangered Taxa of the Italian Flora: *Adenostyles alpina* subsp. *macrocephala* (Asteraceae), as a Case Study

**DOI:** 10.3390/plants12071530

**Published:** 2023-04-01

**Authors:** Valeria Gianguzzi, Giulio Barone, Emilio Di Gristina, Francesco Sottile, Gianniantonio Domina

**Affiliations:** 1Department of Agricultural, Food and Forest Sciences, University of Palermo, Viale delle Scienze, bldg. 4, I-90128 Palermo, Italy; valeria.gianguzzi@unipa.it (V.G.); giulio.barone01@unipa.it (G.B.); gianniantonio.domina@unipa.it (G.D.); 2NBFC, National Biodiversity Future Center, Piazza Marina 61 (c/o palazzo Steri), I-90133 Palermo, Italy; 3Department of Architecture, University of Palermo, Viale delle Scienze, bldg. 14, I-90128 Palermo, Italy; francesco.sottile@unipa.it

**Keywords:** endangered species, ex situ conservation, leaf explant, mountain flora

## Abstract

The conservation of endangered, rare, and endemic plant species is based on in situ and ex situ conservation strategies. When in situ conservation alone is not sufficient to guarantee the survival of the species, ex situ techniques are adopted in support. This study aimed to develop an efficient micropropagation protocol for *Adenostyles* by evaluating the effect of different plant growth regulators on leaf explants. *Adenostyles alpina* subsp. *macrocephala* (Asterace) is a perennial herbaceous plant endemic to Calabria (Southern Italy). The genus *Adenostyles* includes three species confined to the mountains of the Mediterranean and southern Europe. For callus induction, media supplemented with different concentrations of Benzylaminopurine (BAP) (0.5, 1, 2, and 3 mg L^−1^), Naphthaleneacetic Acid (NAA) (1 mg L^−1^), and 2,4-Dichlorophenoxyacetic Acid (2,4-D) (1 mg L^−1^) were tested. Shoot regeneration and proliferation were obtained in media supplemented with BAP (1, 2, and 3 mg L^−1^) and NAA (1 mg L^−1^). Root induction was obtained in media supplemented with IBA (0.25, 0.50, and 1 mg L^−1^) and NAA (0.25, 0.50, and 1 mg L^−1^). Statistically significant differences in callus induction and shoot regeneration were observed between the various media tested. The medium containing Murashige and Skoog (MS) supplemented with 3 mg L^−1^ of BAP and 1 mg L^−1^ of NAA showed the highest percentage of callus induction and increased shoot regeneration. The regenerated shoots showed more effective root induction in the hormone-free MS medium and in the presence of Indole-3-Butyric Acid (IBA) at concentrations of 0.25, 0.50, and 1 mg L^−1^. These results can be used as a basis for the preparation of a micropropagation protocol for different taxa of *Adenostyles*, as well as other species of Asteraceae specialized to the Mediterranean mountain habitat.

## 1. Introduction

The major threats affecting biodiversity in terms of plant distribution and richness include habitat loss and fragmentation, overexploitation, invasive alien species, air pollution and nitrogen deposition, and, finally, climate change [1]. These threats are mostly due to human pressure and demand for resources [2]. Endemic taxa, among all those occurring in a territory, represent one of the greatest global conservation responsibilities for any country [3]. Due to their rarity, they are considered to have low-density occurrences in a small geographic range, which is often a precursor to extinction [4].

There are several possible conservation actions that can be undertaken in situ to help mitigate this negative trend, but when this step is not possible (e.g., loss of habitat), ex situ collections (i.e., outside of the plant’s natural habitat) in botanical gardens, seed banks, in vitro techniques, and cryopreservation collections can be a valuable solution to avoid the complete disappearance of a taxon [5,6].

In situ conservation of threatened and endemic plants is the most effective tool against extinction [7]. This strategy aims to protect and maintain populations within their natural locations. Their habitats are conserved, and single taxa are protected. In situ conservation alone may not always be practical due to several constraints, including reduced area size for target-taxa preservation, low genetic diversity, personnel expenses, and vulnerability to natural uncertainties. [8]. In these cases, ex situ programs can support in situ conservation and, sometimes, are the only conservation strategies possible [9].

Ex situ conservation involves preserving threatened taxa outside of their original habitat by placing them in a human-controlled environment. One of the major limitations is the scarcity of propagating material [10]. Often, we deal with plants that flower occasionally, do not produce fertile seeds, or where traditional vegetative multiplication is difficult to apply. Compared with conventional methods, plant-tissue culture requires a small amount of initial plant material for plant propagation [11]. Among the advantages of in vitro culture techniques are their independence from seasonality, their speed, and their ability to produce pathogen-free plants [12]. On the other hand, the genetic and physiological uniformity of the material obtained can represent a limit with respect to the objective of conservation.

Taxa preserved ex situ as seeds, tissue cultures, or living collections can then be employed for reintroduction, population reinforcement, or restoration actions in situ. There is growing literature about restoration actions, but few studies have reported the outcomes of these projects [13,14]. When selecting a species deserving conservation actions, there are four steps required before any translocation is undertaken: the selection and profiling of the target species, seed collection, development of propagation protocols, and assessment of plant fitness of the populations used as a source [15]. Complementary strategies of ex situ conservation for species with few or no seeds available or those producing recalcitrant ones can be the use of in vitro conservation, which consists of maintaining the plant germplasm using tissue culture technologies [16]. Furthermore, micropropagation techniques can be used to safeguard species that have difficulty being propagated with traditional propagation techniques; for example, *Capparis spinosa* L. (Capparaceae) has a low seed germination rate due to seed dormancy and a high degree of seed heterozygosity. However, the main problems associated with vegetative propagation by stem cuttings seem to depend on the type of propagation material, which is influenced by the environment. For these reasons, in recent years, there has been growing interest in micropropagation with the aim of obtaining a large number of genetically homogeneous and uniform plant materials in a limited time and space [17,18,19]. In addition, micropropagation techniques can also be employed to study the ecological factors that influence plant development [20,21]. Collecting and maintaining endangered plants in tissue culture or cryopreservation to support the species’ survival and reduce its extinction risk can be a valuable alternative [22,23,24,25]. However, these solutions are only feasible if the practitioner can restore a rooted plant from tissue culture or cryopreservation that is then capable of being transplanted into the wild [26,27]; hence, the importance of developing successful protocols arises. Micropropagated plants can present in vitro generated variations, known as somaclonal variation [22,23]; these variations deserve to be monitored, as they can be an advantage for the conservation of wild plants with low genetic variability [25].

The Italian territory lies at the center of one of the main biodiversity hotspots for endemic plants, the Mediterranean basin [28,29]. However, here, as in many Mediterranean countries, very small actions have been undertaken to successfully conserve this plant richness [1]. International treaties (i.e., the Convention on Biological Diversity’s 2020 target) and conservation policies (i.e., Directive 1992/43/EEC in Europe) are in place to reduce biodiversity loss. However, the most effective way to protect biodiversity is at national and local scales [30].

*Adenostyles* Cass. (Asteraceae) is a genus of perennial herbaceous plants endemic to Europe, distributed in the high mountain ranges at montane to subalpine altitudes [31]. This genus includes three species: *Adenostyles alliariae* (Gouan) A. Kern., *A. alpina* (L.) Bluff and Fingerh., and *A. leucophylla* (Willd.) Rchb. *A. alliariae* is further subdivided into two subspecies, and *A*. *alpina* is subdivided into six subspecies [32,33]. This taxonomic arrangement is the result of a spatial and ecological separation between the different populations of the genus that are evolving towards increasingly isolated taxa [33]. Isolated and fragmented populations are most at risk of extinction. The southernmost populations of *Adenostyles* occur in Italy and are those of *A. alpina* subsp. *macrocephala* (Huter, Porta, and Rigo), Dillenb. and Kadereite, and *A. alpina* subsp. *nebrodensis* (Wagenitz and I.Müll.) Greuter. *Adenostyles alpina* subsp. *macrocephala* is endemic to Aspromonte and Sila (Calabria, S. Italy); its populations are scattered but include several hundred mature individuals growing along the rivulets that run through these mountain systems. *Adenostyles alpina* subsp. *nebrodensis* is known from a single locality at Passo della Botte on the higher Madonie mountains in Northern Sicily. The past population’s significant size is evidenced by the abundance of 19th-century specimens found in the main European herbaria. For over twenty years, only one individual has been known to have survived the indiscriminate scientific collection and alteration of its habitat [34]. The Passo della Botte area is, in fact, subject to water collection for civil use, which drains the watercourse and changes the microclimatic conditions of the site. Surely the ongoing climate change, with the decrease in snowfall and the inconstant distribution of rain, plays a role in this modification of microclimatic conditions [35]. These two subspecies have also been investigated from a conservation point of view. The IUCN category endangered (EN) was proposed for *A. alpina* subsp. *macrocephala* (Huter, Porta, and Rigo) Dillenb. and Kadereit, and critically endangered (CR) for *A. alpina* subsp. *nebrodensis* (Wagenitz and I.Müll.) Greuter on the basis of their limited extent of occurrence and area of occupancy and the threats of “Human intrusion & Disturbance” and “Natural System modifications” [36]. In fact, the areas where they grow are affected by the opening of new roads and trails and by water collection. The individual *A. alpina* subsp. *nebrodensis* has been monitored regularly for about 20 years; it flowers occasionally, but no fertile seeds have been observed. Therefore, vegetative reproduction is the only method for obtaining more individuals of this plant, and the micropropagation technique is surely a preferential solution to reduce disturbance [37]. Previous experiments were conducted directly on *A. alpina* subsp. *nebrodensis* have led to the formation of a callus starting from transverse sections of the leaf petiole, but further results have not yet been obtained [38]. The aim of this study was to develop an efficient micropropagation protocol for *Adenostyles,* endangered taxa, by evaluating the effects of different plant growth regulators on leaf explants. We preferred to carry out experimentation on *Adenostyles alpina* subsp. *macrocephala,* given the numerical consistency of its populations.

## 2. Results and Discussion

Leaf segments were cultured on MS medium supplemented with BAP (0.5, 1, 2, and 3 mg L^−1^) in combination with NAA and 2,4-D, respectively, at the same concentration of 1 mg L^−1^. After 3 weeks, there were significant differences (*p* < 0.05) in callus regeneration on media with different combinations and concentrations of cytokinins and auxins. Both in the explants placed directly in the light (photoperiod of 16 h light and 8 h darkness) and those placed in the dark (photoperiod of 24 h darkness) for a week, there was proliferation on all culture media, except in the hormone-free control medium, in which there was no callus production. In the data obtained, there were statistically significant differences between the growth media used, which influenced callus production. The concentration of plant regulators in the medium favored the growth of the callus, which increased when a higher concentration of BAP was present in the medium in combination with 2.4 D. In the present study, both in the case in which the explants were placed directly in the light and dark for one week, the highest percentages of callus production (85% and 66%, respectively) were obtained in the D4 medium (3 mg L^−1^ of BAP and 1 mg L^−1^ of 2,4-D) (Figure 1a,b).

In studies on callus production in *Rubus idaeus* L. (Rosaceae) and *Orthosiphon stamineus* Benth. (Lamiaceae), 2,4-D was the best growth regulator for inducing callus formation [39,40]. In *Dendranthema × grandiflorum* (Ramat.) Kitamura (Asteraceae) leaf explants reported no callus formation in hormone-free media, while vigorous callus growth was observed in the medium with 1.0 mg L^−1^ 2,4-D and 1.0 mg L^−1^ BAP [41]. In several species, somatic embryo induction was achieved using different auxins such as 2,4-D, NAA, IAA, for example *Swietenia macrophylla* King (Meliaceae) with 2,4-D (4.0 mg L^−1^) and kinetin (1.0 mg L^−1^) [42]; *Cymbopogon jwarancusa* (Jones) Schult. (Poaceae) with 2,4-D (18.1 μM) [43]; *Musa acuminata* Colla (Musaceae) with 2,4-D (4.5 μM) [44]; *Salicornia brachiata* Roxb. with 2,4-D (2.0 mg L^−1^) [45], and *Solanum melongena* L. (Solanaceae) with 2,4-D (0.5 mg L^−1^) [46].

In the present study, when 2,4-D was replaced with NAA in the medium, callus production was lower. When the explants were placed in the dark for one week, 15% was recorded in the N1 medium and up to 47% in the N4 medium. In contrast, when the explants were exposed directly to light, 33% were observed in the N1 medium and 60% in the N4 medium. In *Artemisia absinthium* L. (Asteraceae), [47] reported the best callogenic response with BAP and NAA in the medium. A combination of BA and NAA was suitable for callus induction and shoot differentiation. This cytokinin–auxin combination has been used for callus regeneration in various protocols developed for species of *Helianthus* and *Saussurea* (Asteraceae) [48,49,50,51,52,53].

The texture and color of the callus changed according to the presence of growth regulators in the medium. When BAP was present in the medium at a concentration of 1, 2, or 3 mg L^−1^ in combination with 1 mg L^−1^ NAA and the explants were placed in the dark for a week, the callus was yellowish, green, and crumbly. Furthermore, after 5 weeks, small shoots and roots formed in the same medium. However, when the NAA was replaced with 2,4-D, the callus was light green, whitish, and crumbly, and there was shoot formation. The best results in terms of shoot regeneration were obtained in the MS medium integrated with 3 mg L^−1^ BAP and 1 mg L^−1^ NAA. Thus, shoot regeneration was achieved through indirect organogenesis.

Yellowish and whitish soft calli favored the development of shoots (Figure 2).

The mean value of shoots per explant obtained was 3.21 for medium N2, 2.98 for medium N3, and 4.38 for medium N4 (Figure 3). An increase in the vigor of the shoots obtained is therefore observed in the medium with increasing concentrations of cytokinin (BAP) in combination with 1 mg L^−1^ NAA. The culture medium containing 3 mg L^−1^ BAP and 1 mg L^−1^ NAA gave the best results (Figure 4).

Media containing benzyladenine (BA) (2.22–8.87 μM) and NAA (0.27–0.54 μM) were preferably used for multiple shoot induction and maintenance in Asteraceae [54,55,56,57,58,59]. The presence of auxins and cytokinins in the culture medium promotes cell division and differentiation of the shoots [60,61]. In the genus *Artemisia*, the effect of BA at concentrations of 0.1–2 mg L^−1^ on stimulating shoot formation was reported, and the best results occurred at concentrations between 0.25 and 1 mgL^−1^ [62,63].

Hyperhydricity (vitrification) is one of the problems encountered in micropropagation. Vitrified shoots have smaller internodes and poor rooting ability [64]. No hyperhydricity was observed during our tests. This result may be due to the correct concentration of plant growth regulators in the medium, especially cytokinins, whose high concentration has been indicated among the factors that may induce vitrification [65].

To promote root induction, the shoots obtained were transferred to a hormone-free MS medium or a medium containing IBA (0.25, 0.50, and 1 mg L^−1^) or NAA at the same concentrations. After 5 weeks, the shoots grew roots in the medium without hormones and in the presence of IBA in all three concentrations tested. No roots were formed when NAA replaced IBA in the medium. Longer (until 60 mm), more numerous roots (about 10), and more developed shoots (average of 8–10 leaves 30 mm long) were obtained in the hormone-free medium and the medium containing 1 mg L^−1^ IBA. When the concentration of IBA in the medium was 0.25–0.50 mg L^−1^, shorter (about 20 mm) and more numerous roots (about 10) were obtained with less developed shoots (an average of 5 leaves 20 mm long). Among the auxins, IBA is one of the most used to induce root formation by organogenesis. Its activity has been extensively tested on *Arabidopsis* (Brassicaceae) [66,67]. Recent studies conducted on *Stevia rebaudiana* (Bertoni) Bertoni (Asteraceae) have reported that 100% rooting was obtained in MS medium containing 1 mg L^−1^ IBA [68,69].

Subsequently, the microshoots that produced roots were acclimatized by transferring them in conditions of absolute sterility into Jiffy-7^®^ pots soaked with sterile double-distilled water in Magenta^®^ vessel GA7 and placed in the growth chamber under the same thermal (25 ± 1 °C) and light (16 h light and 8 h darkness) phase conditions of the in vitro phase until the emission of roots outside the Jiffy-7^®^ (Figure 5).

The percentage of plants acclimatized to greenhouse conditions was 70%. The individuals produced have a number of leaves between 10 and 15, with an average height of 20 cm. The health status is overall good. Despite the non-optimal transplant period and environmental conditions, the satisfactory results obtained can be attributed to the good quality of the roots produced.

Plants moved outdoors, on hills, and entered vegetative rest after two weeks, which is in accordance with the taxon’s vegetative cycle. This outdoor parcel will be used in the next few years to monitor the acclimatization in natural conditions of these individuals obtained in vitro.

## 3. Materials and Methods

### 3.1. Plant Material and Culture Media

#### 3.1.1. Collection and Sterilization of Plant Material

We opted to use small leaf portions as propagation material due to their ease of harvesting and minimal harm to the plants when cut. The apical/nodal explants are not to be taken from rhizomes that are underground. Young leaves with an area of about 10 cm^2^ of *Adenostyles alpina* subsp. *macrocephala* were collected from different individuals randomly selected within the population growing along the Torrente Telese, Gambarie (South Italy, Calabria), 1420 m a.s.l., 38.175053° N, 15.856755° E (WGS84), on 16 June 2022, by G. Domina, E. Di Gristina, and G. Barone (voucher specimen housed in the SAF herbarium of the Department of Agriculture, Food, and Forest Sciences of the University of Palermo). The leaves were transported to the laboratory in plastic bags stored at about 7 °C and processed within 12 h of collection. The process of sterilizing plant material (explants) is an important activity in tissue culture which aims to eliminate microorganisms carried during explant extraction, which can cause contamination. The leaves were disinfected by standard methods using ethanol/sodium hypochloride [70]. The explants were sterilized by immersion in a 70% ethanol solution for 5 min and then in a 15% sodium hypochlorite solution for 20 min, followed by three 5 min rinses in sterile distilled water. No antibiotics were used. The concentration and exposure time to sterilants were decided based on the consistency of the explants. Subsequently, the leaves were cut in segments of about 50 mm^2^ with a sterile scalpel and cultured in Petri dishes containing mediums medium with different concentrations and combinations of Plant Growth Regulators (PGRs) (Table 1) for 3 weeks.

#### 3.1.2. Callus Induction

For callus induction, the explants were used to inoculate the MS basal medium with vitamins [71], supplemented with 30 g/L^−1^ sucrose and 7 g/L^−1^ plant agar (Duchefa) as a gelling agent, and containing various concentrations and combinations of 6-benzylaminopurine (BAP), 2,4-dichlorophenoxyacetic acid (2,4-D), and 1-naphthylacetic acid (Table 1). MS medium without any hormones (MS0) was used as a control. The pH of the medium was adjusted to 5.5 prior to the addition of the gelling agent with NaCl or with HCl 1 M. Each treatment included 40 explants, or eight replicates (Petri dishes) with five explants each. The callus was subcultured once in the same medium, 10 days after being cultured. After 3 weeks, the effect of the different PGRs on the explants placed directly in the light and those placed in the dark for a week was evaluated. The Petri dishes were maintained in a climatic growth chamber at 25 ± 1 °C in the dark and then placed under a cool, white, fluorescent lamp with a photosynthetic photon flux density (PPFD) of 35 μmol m^−1^ s^−1^ and a photoperiod of 16 h light and 8 h darkness.

#### 3.1.3. Shoot and Root Induction

For shoot induction, calli obtained from the induction phase of callus were transferred into transparent microbox vessels of 125 × 65 mm and 80 mm high, containing 80–100 mL of MS basal medium with vitamins [71], supplemented with 30 g/L^−1^ sucrose and 7 g/L^−1^ plant agar (Duchefa) as a gelling agent, and different concentrations of BAP (1, 2, and 3 mg L^−1^) and NAA (1 mg L^−1^) (Table 1). After 5 weeks, the percentage of shoots produced by the callus was evaluated with binoculars. Subsequently, the 10 mm shoots were isolated from the callus with a sterile scalpel and tweezers. On average, 6 shoots per callus were obtained and subcultured once in the same medium (N2, N3, and N4) for 12 days to promote their elongation. Microshoots, 15 mm long, were used for root induction. The shoots’ proliferation was evaluated only in the media in which there was the development of shoots (N2, N3, and N4).

Rooting of regenerated shoots was achieved on MS medium without any added phytohormones (MS0) or supplementation with different auxins (Table 2). The explants have always been kept on the same medium until the emission of roots without performing subculture. After 5 weeks, once the roots were obtained, the plants were placed in Jiffy-7-Pellet^®^. All cultures were incubated in a climatic growth chamber at 25 ± 1 °C with a photoperiod of 16 h light and 8 h darkness and a photosynthetic photon flux density (PPFD) of 35 μmol m^−1^ s^−1^ for 4 weeks.

#### 3.1.4. Acclimatization

Acclimatization of plantlets was performed first in plastic containers measuring 119 × 90 mm and 80 mm high and containing 150 g of a peat (75%) and perlite (25%) mix under controlled growth conditions (25 ± 1 °C and 16 h light and 8 h darkness) to promote the hardening of plantlets. The mix of peat and perlite used ensured good aeration of the root system, which is considered one of the key elements for the survival of tissue culture plantlets transferred from a lab to pots [72]. A plant rooted in Jiffy-7-Pellet^®^ with a length of 5 cm was placed inside each individual plastic container. After 3 weeks, the plants reached about 15 cm in length and were transplanted in pots of 16 cm in diameter containing about one liter of slightly alkaline loam soil and placed in unheated, shaded greenhouse conditions (10–22 °C and 9 h light and 13 h darkness) for 4 weeks. One plant was transplanted into each pot.

After greenhouse acclimatization, in the first week of December 2022, ten plants were transferred outdoors to study the behavior of in vitro produced plants in climatic conditions more similar to those of the natural population from which the propagating material was taken. It was chosen to grow the plants along a rivulet in a private garden in Giacalone, a locality not far from Palermo (38.033398° N, 13.234257° E, 650 m a.s.l., upper Mesomediterranean bioclimate, on clay carbonate soil). This locality is more than 70 km as the crow flies from the individual of *Adenostyles alpina* subsp. *nebrodensis*, to avoid any risk of genetic pollution.

### 3.2. Data Analysis

Each treatment included 40 explants—eight replicates with five explants each. Statistical analysis was performed using the software SYSTAT 13. To highlight statistically significant differences and possible interactions between the culture system and PGRs, a two-way analysis of variance (ANOVA) was performed (*p* ≤ 0.05). When the interaction between two factors was not significant, a one-way ANOVA was performed; each factor was analyzed individually, and the separation of the averages was performed via the Tukey test (*p* ≤ 0.05).

## 4. Conclusions

Micropropagation, starting from leaf portions of *Adenostyles alpina* subsp. *macrocephala*, was tested with success. The best results with growth regulators were achieved with the following supplements: 3 mg L^−1^ BAP and 1 mg L^−1^ 2.4 D in the initiation stage and 3 mg L^−1^ BAP and 1 mg L^−1^ NAA in the elongation stage. The acclimatization stage was successfully performed with Jiffy-7^®^ pellets. Using this protocol, 12 plantlets can be produced from a 1 cm^2^ leaf in approximately 24 weeks.

To date, no published reports on the use of micropropagation for *Adenostyles* exist. This is also the first report on the outdoor planting of in vitro micropropagated *Adenostyles* species. More years of observations will be needed to monitor the outdoor acclimatization of the plants produced with micropropagation techniques. In particular, it will be necessary to observe whether these individuals are more delicate than those obtained by seed or grown under natural and less favorable conditions.

The proposed micropropagation technique has the potential to be applied to other endangered *Adenostyles* spp. or, more generally, to other Asteraceae of rocky environments, and this in vitro approach may be of benefit to other plant conservation researchers. Although the leaves are not among the best organs from which to start micropropagation [16], they are certainly the easiest to collect and whose cutting causes less stress to the mother plant. Micropropagation from vegetative organs creates clones. Therefore, it is essential for conservation programs to assess, ex ante, the intra-population genetic variability of the taxa for the proper development of conservation strategies and for proper plan sampling [73,74]. Plant regeneration via callus formation often gives rise to somaclonal variations [22]. Variations in plants can be analyzed by various molecular markers or can be estimated by flow cytometry (FCM) [23,24]. Somaclonal variations, when they do not involve a change in the chromosome number, can be useful for increasing the genetic variability of the propagation material to be used for conservation interventions [25].

The first application of the micropropagation protocol presented could be the conservation of *Adenostyles alpina* subsp. *nebrodensis*. The multiplication of this plant must be carried out as soon as possible to offer greater chances of survival to the single individual that still exists.

Micropropagation has not yet been universally adopted for conservation purposes due to still limited knowledge, high production costs, and the need for specialized personnel and equipped laboratories. In addition, the significant loss of material due to microbial contamination, poor rooting, and low survival rates during the acclimatization stage should be taken into account [75,76]. The use of ex situ strategies is expected to rise in view of the increasing anthropogenic pressure and climate change. In vitro techniques for the propagation and preservation of cells, tissues, and plant organs are proving to be a valuable tool in ex situ conservation. As often reiterated, conserving endemic species that are threatened is critical in achieving global biodiversity targets. Each country has the responsibility to halt global biodiversity loss by conserving as a priority the endemic species that occur in its territory. Any loss of endemic wildlife that can no longer be recovered is a failure perpetrated against future generations.

## Figures and Tables

**Figure 1 plants-12-01530-f001:**
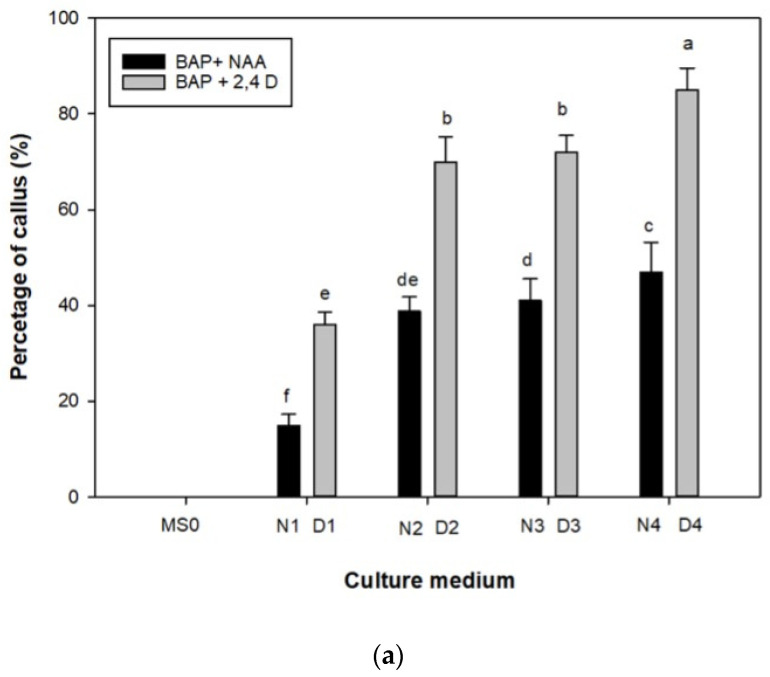
(**a**) Effect of different plant growth regulator (PGR) concentrations on the percentage of callus induction (%) in *Adenostyles alpina* subsp. *macrocephala* placed in the dark for one week (n = 40). Bars represent standard error. The different letters reported above the bars indicate statistically significant differences between each culture medium tested (Tukey’s test, *p* ≤ 0.05). MS0 (control, hormone-free, medium with sucrose); N1 (medium with 0.5 mg L^−1^ BAP + 1 mg L^−1^ NAA); N2 (1 BAP + 1 NAA); N3 (2 BAP + 1 NAA); N4 (3 BAP + 1 NAA); D1 (medium with 0.5 mg L^−1^ BAP + 1 mg L^−1^ 2,4-D); D2 (1 BAP + 1 2,4-D); D3 (2 BAP + 1 2,4-D); D4 (3 BAP + 1 2,4-D). (**b**) Effect of different plant growth regulator (PGR) concentrations on the percentage of callus (%) in *Adenostyles alpina* subsp. *macrocephala* placed directly in the light (n = 40). Bars represent standard error. The different letters reported above the bars indicate statistically significant differences between each culture medium tested; (Tukey’s test, *p* ≤ 0.05). MS0 (control, hormone-free, medium with sucrose); N1 (medium with 0.5 mg L^−1^ BAP + 1 mg L^−1^ NAA); N2 (1 BAP + 1 NAA); N3 (2 BAP + 1 NAA); N4 (3 BAP + 1 NAA); D1 (medium with 0.5 mg L^−1^ BAP + 1 mg L^−1^ 2,4-D); D2 (1 BAP + 1 2,4-D); D3 (2 BAP + 1 2,4-D); D4 (3 BAP + 1 2,4-D).

**Figure 2 plants-12-01530-f002:**
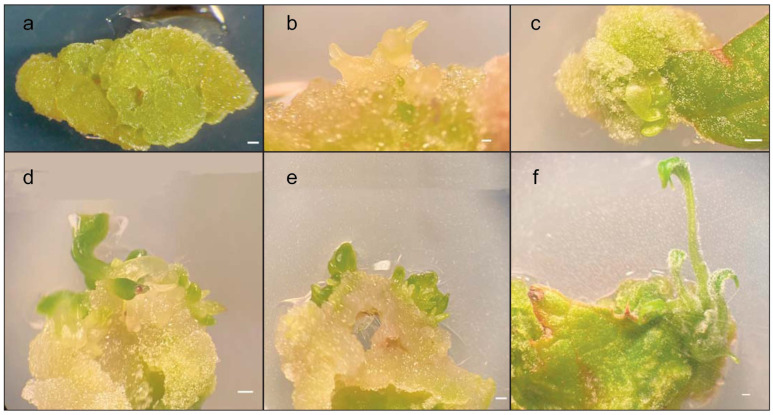
In vitro callus regeneration produced by leaves of *Adenostyles alpina* subsp. *macrocephala*. (**a**) The callus that developed in the medium when the NAA was replaced with 2.4 D. (**b**–**f**) Different stages of indirect organogenesis in the MS medium integrated with 3 mg L^−1^ BAP and 1 mg L^−1^ NAA. Bars (**a**–**f**) = 1 mm.

**Figure 3 plants-12-01530-f003:**
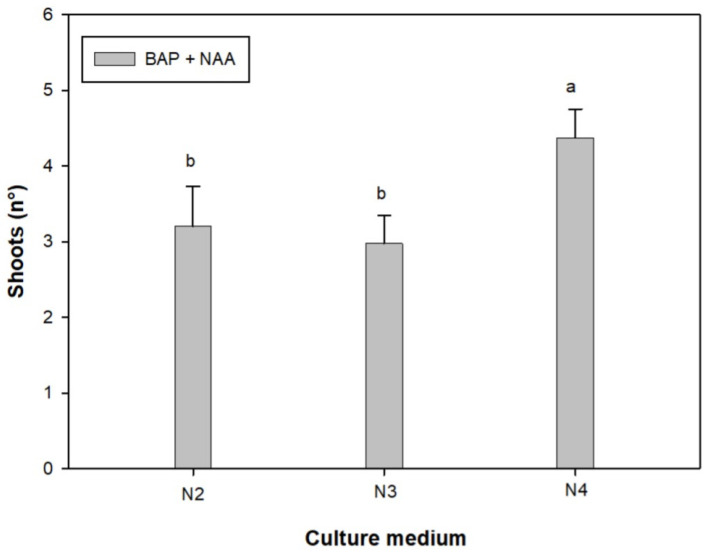
Effect of different PGR concentrations on the number of shoots per explant. Bars represent standard error. The different letters reported above the bars indicate statistically significant differences between each culture medium tested (Tukey’s test, *p* ≤ 0.05). N2 (medium with 1 mg L^−1^ BAP + 1 mg L^−1^ NAA); N3 (2 BAP + 1 NAA); N4 (3 BAP + 1 NAA).

**Figure 4 plants-12-01530-f004:**
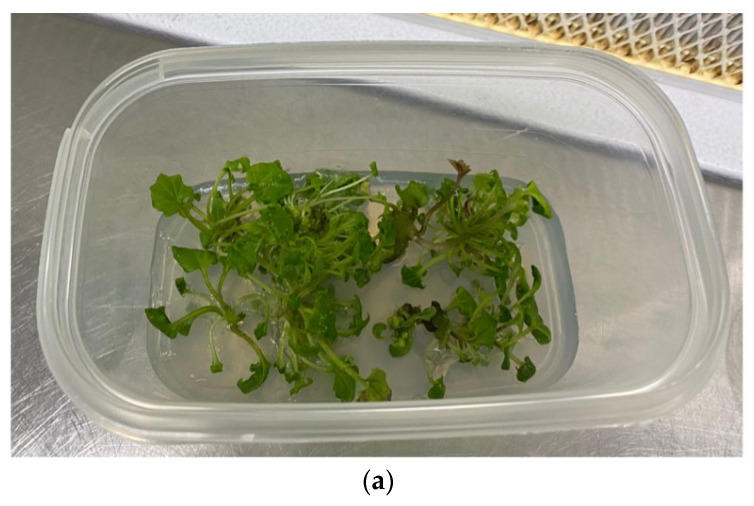
In vitro shoot development of *Adenostyles alpina* subsp. *macrocephala* in the culture medium containing 3 mg L^−1^ BAP and 1 mg L^−1^ NAA (**a**,**b**).

**Figure 5 plants-12-01530-f005:**
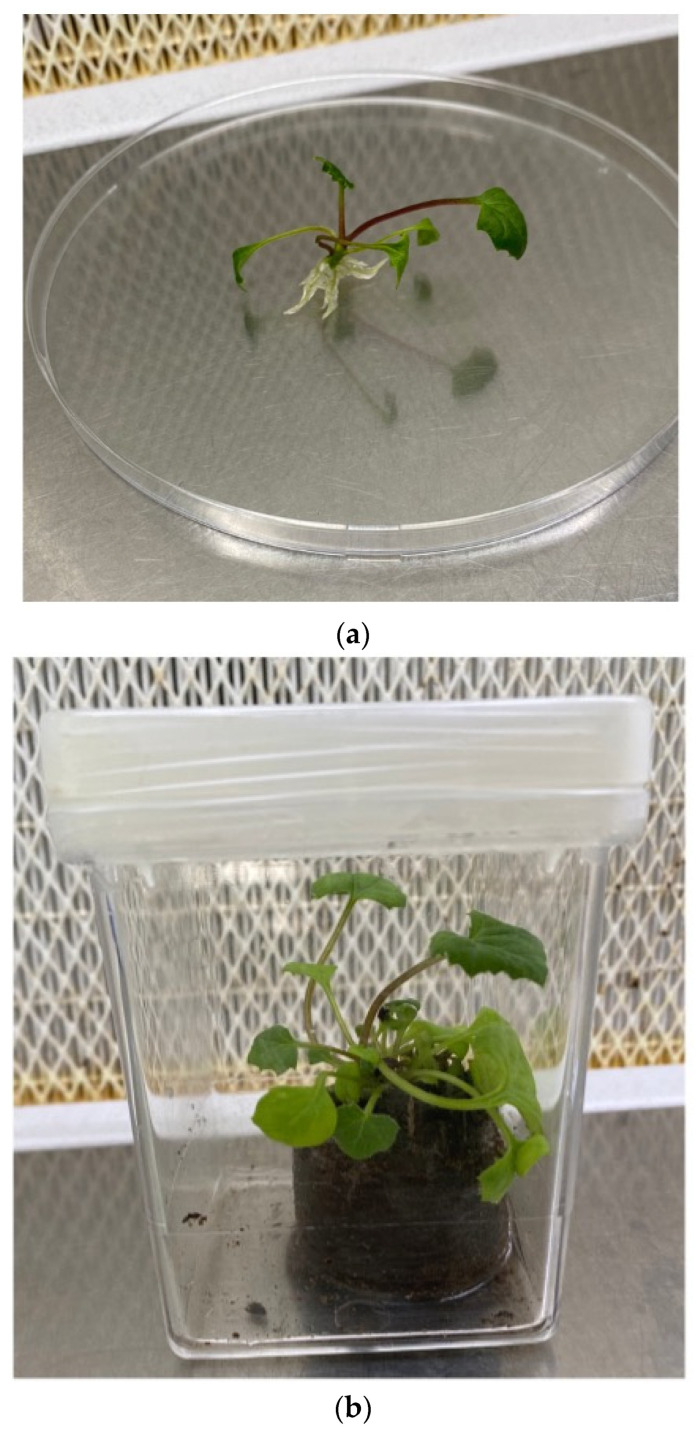
*Adenostyles alpina* subsp. *macrocephala*: (**a**) In vitro rooting of shoots. (**b**) Plantlet placed in Jiffy-7^®^. (**c**) Plant in greenhouse for acclimatization.

**Table 1 plants-12-01530-t001:** Plant growth regulators used in the growth media to induce the formation of callus and shoots from *Adenostyles alpina* subsp. *macrocephala* leaves.

Medium	PGR		
	BAP mg L^−1^	NAA mg L^−1^	2,4-D mg L^−1^
MS0	0	0	0
N1	0.5	1	0
N2	1	1	0
N3	2	1	0
N4	3	1	0
D1	0.5	0	1
D2	1	0	1
D3	2	0	1
D4	3	0	1

Abbreviations: MS0 (control, hormone-free, medium with sucrose); N1, (medium with 0.5 mg L^−1^ BAP + 1 mg L^−1^ NAA); N2 (1 BAP + 1 NAA); N3 (2 BAP + 1 NAA); N4 (3 BAP + 1 NAA); D1 (medium with 0.5 mg L^−1^ BAP + 1 mg L^−1^ 2,4-D); D2 (1 BAP + 1 2,4-D); D3 (2 BAP + 1 2,4-D); D4 (3 BAP + 1 2,4-D).

**Table 2 plants-12-01530-t002:** Plant growth regulators used in the growth media to induce the formation of roots from *Adenostyles alpina* subsp. *macrocephala* leaves.

Medium	PGR	
	IBA mg L^−1^	NAA mg L^−1^
MS0	0	0
R1	0.25	0
R2	0.50	0
R3	1	0
R4	0	0.25
R5	0	0.50
R6	0	1

Abbreviations: MS0 (control, hormone-free, medium with sucrose); R1, R2, R3, (medium with different concentrations of IBA, 0.25 mg L^−1^, 0.50 mg L^−1^, 1 mg L^−1^, respectively); R4, R5, R6 (medium with different concentrations of NAA, 0.25 mg L^−1^, 0.50 mg L^−1^, 1 mg L^−1^, respectively).

## Data Availability

The data presented in this study are available in the text.

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
