# Peer review of "Micropropagation of Endemic Endangered Taxa of the Italian Flora: Adenostyles alpina subsp. macrocephala (Asteraceae), as a Case Study"

_plants, 2023, doi:10.3390/plants12071530_

Round 1

Reviewer 1 Report

The manuscripts describes micropropagation of a endangered endemic  species of  Adenostyles Cass. (Asteraceae).  The manuscript  can be improved further as results can be better presented. The interpretations lack evidence and clarity---

(1) Since micropropgation was the aim, why authors preferred a callus mediated shoot regeneration  and not propagate through shoot tip/ nodal explants to maintain genetic fidelity of the TC plants ? 

Fig 1--% callus induction --mention n= ?

Line -280- and Fig 2 -- There is no evidence for somatic embryogenesis. 

Fig 3----Unclear  

Authors have made confusing statements , results shown indicate callus mediated  organogenesis  in the species.

Author Response

See attached letter

Reviewer 2 Report

I reviewed the paper titled Micropropagation of endemic endangered taxa of the Italian Flora. Adenostyles alpina subsp. macrocephala (Asteraceae), as a case study. The objective of this study is to develop an efficient micropropagation protocol for Adenostyles endangered taxa, evaluating the effect of different plant growth regulators on leaf explants. This article is valuable. Only the following corrections should be made.

Abstract

The treatments used in the experiment should be stated in more details and clearly. For example, the type and amount of plant growth regulators used should be mentioned.

Keywords

Keywords should be written in alphabetical order.

Introduction

It is recommended to do a better literature review and address to the relevant papers. For example the authors can address to the following literatures:

https://doi.org/10.17660/ActaHortic.2005.705.24

https://doi.org/10.1016/j.scienta.2017.03.045

https://doi.org/10.1016/j.scienta.2017.04.023

Material and Methodology

2.1.1. Collection and sterilization of plant material

Please mention in this section at the end line what sizes of the sterilized leaves were cut and what kind of sterile nutrient medium were used and how long were cultured on the media.

2.1.2. Callus induction

Please mention how often the explants were subcultured.

2.1.3. Shoot and root induction

Please mention the size and amount of shoots separated from the callus and cultivated in the rooting medium.

Please mention how often the explants were subcultured in shooting and rooting medium.

2.1.4. Acclimatization

Please replace Acclimatisation with Acclimatization.

Please mention the size of plantlets cultured in plastic containers and pots.

Please mention the number of plantlets cultured in each plastic container and pot.

Please mention the size of plastic containers and pots.

Please mention the amount of culture medium used in each plastic container and pot.

3. Data analysis

Please mention the number of treatments, replicates and explants cultured in each replicate for each part of the experiment.

4. Results and discussion

In Figure 1, the difference between the graph a and b should also be noted in itself.

In Figure 2, there is no explanation about the Figure e.

In Figure 4, the Figure a and b is not specified.

In the results and discussion section, it is better to use more and up-to-date sources. For example, please see the above-mentioned literature.

Author Response

See attached letter

Round 2

Reviewer 1 Report

I have edited some parts of the manuscript and one figure . Minor corrections required .

Author Response

Dear Reviewer,

thanks so much for your additional suggestions edited in the MS. Accordingly to those suggestions, we modified the text to accept them entirely.

Thanks again for helping us to improve the quality of the MS.

Regards.

the Authors